# Nitric Oxide Distribution Correlates with Intraluminal Thrombus in Abdominal Aortic Aneurysm: A Computational Study

**DOI:** 10.3390/bioengineering12020191

**Published:** 2025-02-17

**Authors:** Siting Li, Shiyi Yang, Xiaoning Sun, Tianxiang Ma, Yuehong Zheng, Xiao Liu

**Affiliations:** 1Department of Vascular Surgery, Peking Union Medical College Hospital, Chinese Academy of Medical Sciences and Peking Union Medical College, Beijing 100730, China; lisitingpaul@126.com (S.L.);; 2State Key Laboratory of Complex Severe and Rare Diseases, Peking Union Medical College Hospital, Chinese Academy of Medical Science and Peking Union Medical College, Beijing 100005, China; 3Key Laboratory of Biomechanics and Mechanobiology (Beihang University), Ministry of Education, Key Laboratory of Innovation and Transformation of Advanced Medical Devices, Ministry of Industry and Information Technology, National Medical Innovation Platform for Industry-Education Integration in Advanced Medical Devices (Interdiscipline of Medicine and Engineering), School of Biological Science and Medical Engineering, Beihang University, Beijing 100191, China; buaa_ysy@buaa.edu.cn (S.Y.);

**Keywords:** abdominal aortic aneurysm, intraluminal thrombus, nitric oxide, computational fluid dynamics

## Abstract

Intraluminal thrombus (ILT) in the abdominal aortic aneurysm (AAA) is associated with disease progression and complications. This study investigates the relationship between nitric oxide (NO) concentration and ILT in AAA patients using patient-specific computational fluid dynamics (CFD) models. Four AAA patients with ILT were enrolled. Patient-specific models of the aorta and branch arteries were constructed followed by CFD simulations. NO concentration was modeled based on endothelial shear stress response and its transport within the arterial lumen and wall. Hemodynamic parameters, including wall shear stress (WSS) and its derivatives, were analyzed alongside NO distribution. ILT accumulation was primarily located in the infrarenal abdominal aorta. Regions of decreased NO concentration correlated with ILT accumulated areas, whereas regions with decreased TAWSS and increased OSI were less consistent with ILT accumulation. A negative correlation was observed between the thrombus area and NO concentration, with *p* values of less than 0.001 for four patients. The time-average area NO concentration values of lumen area with ILT were lower than those of non-ILT sections. Spatially, NO was unevenly distributed, with thicker thrombus in regions of lower NO concentration. NO distribution could serve as a better potential personalized marker for thrombosis prediction in AAA compared to WSS-derived parameters.

## 1. Introduction

Abdominal aortic aneurysm (AAA) is caused by multiple factors which leads to degenerative changes in the abdominal aortic wall structure, abnormal stress, and permanent expansion of the arterial wall. The diagnosis and treatment of ruptured AAA are challenging and associated with a high mortality rate [1]. Intraluminal thrombus (ILT) formation is observed in approximately 75% of AAA patients through imaging examinations [2]. Studies have found that the distribution is associated with the progression and rupture of AAA, and thrombus detachment poses a risk of causing acute lower extremity arterial embolism [3]. In addition, patients receiving endovascular treatment of AAA are also at risk of complications, such as in-stent thrombosis and occlusion following stent graft implantation [4,5]. Individualized thrombosis assessment and prediction for patients would assist in understanding the progression and outcome of AAA and provide guidance for improving the efficacy of surgical intervention and reducing complications.

In recent years, nitric oxide (NO) has attracted increasing attention as a key signaling molecule in the cardiovascular system [6]. Under physiological conditions, NO released by endothelial cells diffuses to the blood vessel wall and relaxes smooth muscle, regulating blood flow and vascular tone [7]. It also plays a critical role in preventing platelet aggregation and adhesion [8]. NO sustained-release graft systems, as a method to reduce the risk of in-stent thrombosis, have also be increasingly studied in recent years [9]. On the other hand, excessive or low NO under pathological conditions may also be harmful to tissues, leading to vascular inflammation, oxidative damage, atherosclerotic plaque rupture, thrombosis, etc. [10,11]. Regarding vascular diseases, a previous study found that NO distribution on the endothelial surface was related to atherosclerotic plaque in carotid artery stenosis [12]. Distinct NO distribution patterns have also been observed in thoracic aortic aneurysm and dissection (TAAD) [13,14]. Furthermore, the disruption of vascular endothelial cells is a key factor contributing to intravascular thrombosis [15]. Thus, we hypothesize that the abnormal distribution of NO in the aorta may contribute to ILT accumulation in AAA.

Since NO produced in blood vessels has the characteristics of rapid diffusion and rapid consumption [16], direct detection of NO concentration in the human body is currently very difficult. Due to the complexity of blood components, blood flow status, and vascular tissue characteristics, establishing a basic molecular biology research model for thrombosis is also challenging [17]. Computational fluid dynamics (CFD) can utilize computer digital modeling and computational simulation to comprehensively explore the interaction of multiple factors, including blood pressure, coagulation factors, and vascular status [18]. CFD could offer the advantage of simulating the production and distribution of NO in vitro. Studies have shown that NO is closely related to the blood flow field. Endothelial nitric oxide synthase (eNOS) can respond to wall shear stress (WSS) and produce NO, with changes in WSS caused by disturbed blood flow leading to uneven NO distribution on the aortic wall surfaces [19]. In previous studies, the NO transport numerical model developed by Liu and colleagues has been used to explore its role in atherosclerosis formation and stent design [12,20]. This study aims to explore whether NO could be used as a better indicator of thrombosis compared to WSS-derived hemodynamic parameters. The concentration and distribution of NO were numerically simulated using the NO transport model to analyze their correlation with patients’ actual ILT status.

## 2. Materials and Methods

### 2.1. Patients and Models

This study enrolled 4 AAA patients with ILT who were admitted to the Department of Vascular Surgery from Peking Union Medical College Hospital. The study followed the Helsinki Declaration and was approved by the hospital’s ethical committee. Written informed consent was obtained from all patients. Preoperative computed tomography angiography (CTA) data of the abdominal aorta and preoperative ultrasound of the branch arteries were collected. The CTA images were first loaded into Mimics (v21.0, Materialize). The threshold segmentation method was used to extract artery geometries of the aorta and visceral branches (celiac trunk, superior mesenteric artery, renal arteries, and the common iliac arteries).

The outer wall of the aorta was used as the model boundaries to simulate the model of the initial state of aneurysm formation without thrombus formation. Since the CT values of the wall and the mural thrombus are not significantly enhanced compared with the surrounding tissues, the corresponding area of the patient’s aorta was segmented using CRIMSON (v2023.06.03) [21]. The patient’s CTA image was imported into the software, and a smooth aortic outer wall model was made by drawing ellipses along the vessel centerlines and lofting. The segmental model was imported into Mimics and combined with the original model to complete model construction. Cross sections of the infrarenal aorta were selected every 5 mm and divided into thrombosis (ILT) and non-thrombosis (non-ILT) areas (Figure 1A). The area of thrombus formation in each section in the ILT group was calculated. The thickness of thrombus was projected onto the outer wall.

The new geometries and their centerlines were then exported into Solidworks (v20, Solidworks Corp.), and the lumen and outer wall of each geometry were separately constructed by circle lofting according to the imported geometry. The thickness of the vessel wall was set as 2 mm for the aorta and 0.75 mm for the visceral branches, and the vessel wall geometries were created with Boolean subtraction (Figure 1B). Both the lumen and vessel geometries were assembled and loaded into FLUENT (v20, Solidworks Corp, Waltham, MA, USA) for further simulation.

### 2.2. Boundary Conditions

For boundary conditions, velocity waveforms were acquired from ultrasound images for the aortic inlet and the branch outlets. All waveforms were smoothed and fitted using the Fourier function in MATLAB (v2021b, MathWorks, Natick, MA, USA) and loaded into FLUENT. The velocity waveforms of the branch outlets are converted into a flow ratio through multiplying it by the outlet cross-sectional area and blood density. Additionally, the arterial wall was assumed to be a rigid-wall no-slip condition.

### 2.3. Meshing

ICEM (v21.0, ANSYS, Inc., Canonsburg, PA, USA) was used to produce the hexahedral mesh using blocking and O-grid techniques, considering that high-quality hexahedral meshes near the wall can more accurately capture boundaries and make the NO concentration distribution smoother. Under steady-state conditions, it is considered that mesh independence has been achieved when the average difference in NO concentration between two consecutive simulations with 10% grid refinement is less than 5% (Appendix A).

### 2.4. Calculation Procedure

The numerical calculations were carried out using Fluent (v21.0, ANSYS, Inc., Canonsburg, PA, USA), a validated finite volume-based algorithm. The mass transport of NO was simulated by a user-defined function (UDF), which was verified by Liu’s previous work [22]. Specifically, general macros were used to define the wall NO generation rate modulated by WSS, as well as the diffusion coefficient and reactions in the lumen and arterial wall, as described in Equations (4)–(8). The WSS was first calculated through flow field simulations and then incorporated into the NO transport calculations. All transient calculations were performed for five cycles, and the result of the fifth cycle was taken as the final simulation result.

### 2.5. Governing Equations

Flow equations:

Blood flow in the arterial lumen is controlled by the Navier–Stokes equation and the continuity equation:(1)$$\rho_{b}(\frac{\partial u}{\partial t}+u\cdot \nabla u)=-\nabla p+\mu_{b}\nabla^{2}u$$(2)∇⋅u=0

Blood was modeled as an incompressible, Newtonian fluid with a density ($\rho_{b}$) of 1056 kg/m^3^ and viscosity ($\mu_{b}$) of 0.0035 Pa·s.

Dynamics of Nitric Oxide Transport:

The simulation of NO transport is governed by the convection–diffusion–reaction equations shown below. In the vessel lumen, the transport of NO in blood can be described by the following equation:(3)$$\frac{\partial c_{l}}{\partial t}+\nabla \cdot \left(-D_{l}\nabla c_{l}\right)+u_{l}\cdot \nabla c_{l}-R_{l}=0$$
where $c_{l}$ is the NO concentration in the blood, $D_{l}$ is the diffusion coefficient of NO in blood and taken as 3.3 × 10^−9^ m^2^/s, and $R_{l}$ represents the reaction [23]. The reaction term here comprises oxidation by oxygen and consumption by red blood cells (RBCs), which is described by pseudo-second-order and first-order reactions, respectively:(4)$$R_{l}=k_{oxgen}c_{l}^{2}+k_{ery}c_{l}\;$$
where $k_{oxgen}$ is the pseudo-second-order reaction rate (7.56 × 10^−6^ nM/s) [24], and $k_{ery}$ is the first-order reaction rate for NO consumption by RBCs [25] in the lumen(2.3 s^−1^).

For the endothelial cells, NO was produced by the cells via eNOS in response to and modulated by wall shear stress [26,27]. The NO production rate (*R_NO_*) of endothelium is assumed to be controlled by the hyperbolic model based on experimental measurements reported by Andrews et al. [28]:(5)$$R_{NO–hyp}=R_{basal}+R_{max}\frac{\left|\tau_{w}\right|}{|\tau_{w}|+b}\;$$
where ***τ_w_*** is the wall shear stress,$\;R_{basal}$ = 2.13 nM/s, $R_{max}$ = 457.5 nM/s, and b = 3.5 Pa to ensure that the amount of NO is at the physiological nM level.

For the arterial wall, due to the low transmural fluid velocity and the high diffusivity of NO in the arterial wall, the convection term of NO was ignored [22]. NO transport in artery walls is characterized by the following equations:(6)$$\frac{\partial c_{w}}{\partial t}+\nabla \cdot \left(-D_{w}\nabla c_{w}\right)-R_{w}=0$$(7)$$R_{w}=k_{w}c_{w}\;$$
where $c_{w}$ is the concentration of NO in the arterial wall, $D_{w}$ is the diffusivity of NO in the arterial wall and equal to 8.48 × 10^−10^ m^2^/s [29], and $R_{w}$ is the reaction of NO, which is treated as a first-order rate expression [30]:(8)$$R_{w}=k_{w}c_{w}\;$$
where $k_{w}$ is the consumption rate constant and assumed as 0.01 s^−1^.

### 2.6. NO Concentration and ILT

Regarding the NO distribution results, the surface-averaged time-averaged concentration of NO (TAcNO) was calculated for the infra-renal lumen surface with and without ILT, as well as for the aforementioned cross-sections taken every 5 mm (Figure 1C).

Several hemodynamic variables and indicators of thrombosis, including time-averaged WSS (time-averaged wall shear stress, TAWSS), oscillatory shear index (OSI), relative residence time (RRT), endothelial cell activation potential (ECAP), and time-averaged spatial WSS gradient (WSSG), were calculated:(9)$$TAWSS=\frac{1}{T}\int_{0}^{T}\left|\tau_{w}\right|dt\;$$(10)$$OSI=\frac{1}{2}(1-\frac{\left|\int_{0}^{T}\tau_{w}dt\right|}{\int_{0}^{T}\left|\tau_{w}\right|dt})$$(11)$$RRT=\frac{1}{(1-2OSI)TAWSS}\;$$(12)$$ECAP=\frac{OSI}{TAWSS}\;$$(13)$$TAWSSG=\frac{1}{T}\int_{0}^{T}\sqrt{{\left(\frac{\partial \tau_{w}}{\partial x}\right)}^{2}+{\left(\frac{\partial \tau_{w}}{\partial y}\right)}^{2}+{\left(\frac{\partial \tau_{w}}{\partial z}\right)}^{2}}dt\;$$

The time-averaged concentration of NO (TAcNO) and other parameters were calculated and illustrated.

In order to explore the relationship between the spatial distribution of NO and thrombosis, three representative sections were taken from the ILT area of the infrarenal abdominal aorta of each patient, 20 mm apart. On the sections, four straight lines were taken from the anterior (a), posterior (p), inner (i), and outer (o) directions. At the same time, the thickness of the thrombus in the four directions of the corresponding cross-section was calculated in Mimics, and the 2 directions with thickest and thinnest thrombus of each section were calculated for the line-averaged TAcNO (Figure 1D).

### 2.7. Statistical Analysis

The least-squares method was used to fit a straight line and calculate the correlation coefficient R^2^ for the surface-averaged or line-averaged TAcNO and ILT area, and the F test was used to verify the goodness of fit. All statistical analyses were performed using GraphPad Prism (v8.0.1, Dotmatics Corp., Boston, MA, USA), and *p* ≤ 0.05 was considered statistically significant.

## 3. Results

### 3.1. Baseline Information

As shown in Table 1, the study included three male and one female patient, aged 63 to 84 years. All patients had cardiovascular risk factors, such as smoking, coronary heart disease, and hyperlipidemia, along with mild renal impairment. All patients were on aspirin. They met the criteria for surgical intervention, successfully underwent endovascular aneurysm repair (EVAR) under general anesthesia, and were discharged without complications. Follow-up imaging at 3 months, 6 months, and 1 year showed no thrombotic complications.

### 3.2. NO Distribution and Thrombosis Indicators

Figure 2 shows the actual condition of thrombosis in the models of the four patients, along with NO concentration and hemodynamic parameters such as TAWSS on the vessel wall. In all four patients, the ILT accumulation area was mainly located in the infrarenal abdominal aorta. Regions with increased wall NO concentration were primarily at the origins of aortic branch vessels, while regions with decreased NO concentration were mainly in the dilated infrarenal abdominal aorta, which was consistent with the ILT accumulation area. In comparison, the regions of decreased WSS or WSSG and increased OSI were less consistent with ILT accumulation. The regions with increased levels of RRT and ECAP, which characterized thrombosis, also correlated with the actual condition of ILT accumulation to some degree, but was not as strongly correlated as the regions of decreased NO concentration and did not encompass all thrombus areas.

### 3.3. NO and Actual ILT Condition

Figure 3 showed the NO concentration at peak systole and ILT thickness projected onto the vessel outer wall. At the time point, the level of NO concentration reached its first peak, and the gradient of NO could be clearly observed. The decreased area of NO concentration correlated well with increased ILT thickness in all four patients. In addition, the curvature of the AAA models is presented (Appendix A).

As presented in Figure 4, the scatter plot shows a negative correlation between thrombus area and the surface-averaged TAcNO values of the ILT cross-sections. As sections were taken from top to bottom in the spatial model, the thrombus area first increased and then decreased, mirroring changes in aneurysm diameter. The fitting straight lines showed that the correlation coefficients (R^2^) of patient No. 1 was 0.8038, the R^2^ of patient No. 2 was 0.8056, the R^2^ of patient No. 3 was 0.8457, and the R^2^ of patient No. 4 was 0.6592, all statistically significant (*p* < 0.001). Figure 5A illustrates that the surface-averaged TAcNO was lower for the infra-renal lumen surface with ILT compared to lumen surface without ILT in all four patients.

### 3.4. Spatial NO Distribution and Thrombus Thickness

Figure 6 shows three representative sections taken in the ILT area of the infrarenal abdominal aorta of each patient. Overall, the NO concentration in the space was unevenly distributed in different directions in the anterior, posterior, inter, and outer sides of the aorta. The thrombus was thinner in directions with higher NO concentration and thicker in directions with lower NO concentration. The average results of three representative sections also showed that the line-averaged TAcNO in the direction of thinnest thrombus was higher than that in the direction of thickest thrombus in all four patients (Figure 5B).

## 4. Discussion

Individualized CFD analysis based on patient-specific models and boundary conditions can be a non-invasive method to simulate disease progression and predict complications. Previous CFD research suggests that thrombosis is related to the distribution of vessel WSS in the flow field, which can be mainly divided into two mechanisms: abnormal activation of platelets under high shear stress and blood flow stasis and accumulation of coagulation factors under low shear stress [31,32]. Based on WSS, researchers have explored various relationships between hemodynamic parameters and thrombosis in AAA. For instance, Himburg et al. studied the effect of the OSI on endothelial cell permeability, and found that appropriate changes in the OSI could better reflect the residence time of blood substances in the local vessel wall, and thus proposed the parameter of RRT [33]. Achille et al. found that OSI could be divided by the TAWSS to indicate the tendency of local thrombosis, defining it as ECAP [34]. However, these parameters have not been verified in a large clinical cohort. Furthermore, in clinical AAA simulations, hemodynamic parameters like RRT and ECAP do not always correspond to the entire area of abdominal thrombosis, and there is still a lot of room for exploration in using blood flow simulation to explore AAA thrombosis. In healthy vascular conditions, NO produced by endothelial cells could achieve thrombotic homeostasis by preventing platelet activation [35,36]. NO can inhibit platelet activation by binding to intracellular soluble guanylate cyclase and activating protein kinase G, or stimulate adenylate cyclase and inhibit platelet aggregation through protein kinase A [37]. It can also reduce the expression of glycoproteins on the platelet membrane surface and inhibit adhesion to the endothelial cell surface [38]. Thus, this study investigated the NO level in AAA and its relationship with ILT. The results showed that the mural distribution of thrombosis of AAA was related to the reduction in NO in the artery.

Prior CFD analyses on the correlation between NO and the cardiovascular system focus mainly on atherosclerotic diseases. The eNOS content in the area where atherosclerotic plaques were generated was significantly lower than that in normal tissue, and the NO production rate was reduced [39]. Arterial stenosis after plaque formation also significantly affected the downstream NO transport [22], and the various components in plaques, such as lipids and blood clots, could also change the production and distribution of NO [12]. Moshfegh et al. used circuit components to simulate and calculate the concentrations of calcium ions and NO in coronary arteries and found that the abnormal distribution of NO was also closely related to vascular damage [40]. In addition, perivascular adipose tissue (PVAT) could also affect NO transport [41].

In terms of dilated lesions, the early application of the present NO transport CFD model in the thoracic aorta showed that NO was uneven in different directions in space and may be affected by the local flow field [19]. Turbulence occurred in the inner side of the thoracic aorta with decreased WSS, and the corresponding NO concentration was also lower than in other directions. In addition, the distribution of NO was affected by the consumption in the vessel lumen flow [19]. A research team from the Netherlands analyzed the distribution of NO in the thoracic aortic aneurysm (TAA) and found that the distribution of NO in TAA was also different from that in the normal thoracic aorta [14]. Our research group previously studied the distribution of NO in a series of visceral aneurysms and found that the aneurysm formation site had abnormal NO distribution compared with the surrounding areas (unpublished). However, there have been no CFD studies reported on the correlation between NO and intra-abdominal aortic thrombosis.

In the model used here, NO was produced by the endothelial cells in response to shear stress and transported into both the vessel lumen and wall. Within the lumen, NO was transported through a combination of diffusion and convection, while being consumed by red blood cells (RBCs) via complex formation with hemoglobin and oxidized to nitrate by oxygen. Consequently, both local WSS and overall flow field condition influence NO distribution, and NO may reflect both endothelial function and flow alterations caused by thrombus accumulation. In this study, the calculated NO concentration of all models was within the reported physiological ranges [42]. The results show that the reduction in NO in AAA was closely related to the actual thrombosis in patients, the cross-sectional NO concentration was negatively correlated with the amount of thrombus formation, and the direction of spatial NO reduction also tended to have thicker thrombus accumulation. The distribution of NO was more consistent with thrombotic area compared to hemodynamic parameters such as WSS/WSSG, RRT, and ECAP. The areas where WSS decreased and RRT and ECAP increased were mainly in the direction of aneurysm expansion, while areas where NO decreased and thick thrombus accumulation areas were not necessarily in the direction of maximum expansion. Therefore, NO may have better potential application value as a personalized thrombosis indicator.

In addition, the correlation analysis between the thrombus area and NO concentration in this study shows that, in patient models No. 2 and 4, as the thrombus area decreased, the NO concentration increased to a lesser extent in the area near the iliac branch than on the aortic area near the renal artery. Both of these two patients had long-expanded volumes of AAA models, and the curvature of the aneurysm at the iliac bifurcation was significantly reduced. Therefore, it is speculated that local thrombosis may be simultaneously affected by geometric conditions, which is consistent with the report from Bhagavan et al. that the more axially distal-located AAA may be related to increased ILT [43]. This result suggests that the actual thrombotic condition in AAA is influenced by multiple factors.

This study has some limitations. The sample size is relatively small due to the time-consuming simulation process. The result could not be validated by direct invasive measurement of NO. Since all patients who came to the hospital already had AAA with ILT on image examination, the outer wall was considered as the initial state of aneurysm formation without ILT, which may not be entirely accurate. NO was only assumed to be produced by the endothelial cells, while iNOS production from the vascular pathological state was not included in the model. In follow-up studies, the sample size should be expanded to verify the negative correlation between NO and thrombosis. Additionally, we hope that an integrated model could be built on this basis to simultaneously analyze multiple factors, such as the geometric conditions of thrombosis, material transport, and boundary conditions to carry out more accurate thrombosis prediction, which could also be extended to visceral and lower limb arteries.

## 5. Conclusions

In conclusion, the reduction in NO in the abdominal aorta was closely related to ILT accumulation in AAA patients. The cross-sectional NO concentration was negatively correlated with the amount of thrombus formation, and areas of spatial NO reduction tend to have thicker thrombus accumulation.

## Figures and Tables

**Figure 1 bioengineering-12-00191-f001:**
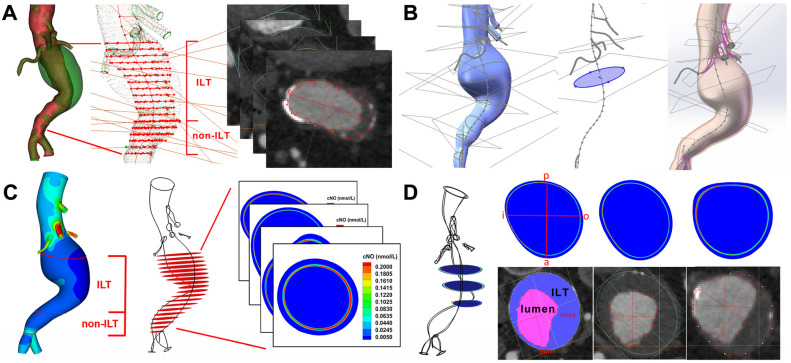
Study flowchart. (**A**) Cross sections of the infrarenal aorta were divided into ILT and non-ILT groups, and the area of sections were calculated. (**B**) The lumen and outer wall of each geometry were separately constructed. (**C**) The NO surface-averaged NO concentration for both ILT and non-ILT lumen areas, as well as that of cross-sectional sections were calculated. (**D**) Spatial distribution of NO (the line-averaged NO of direction with thinnest and thickest thrombus were calculated) and its relationship with ILT was calculated in 3 representative sections.

**Figure 2 bioengineering-12-00191-f002:**
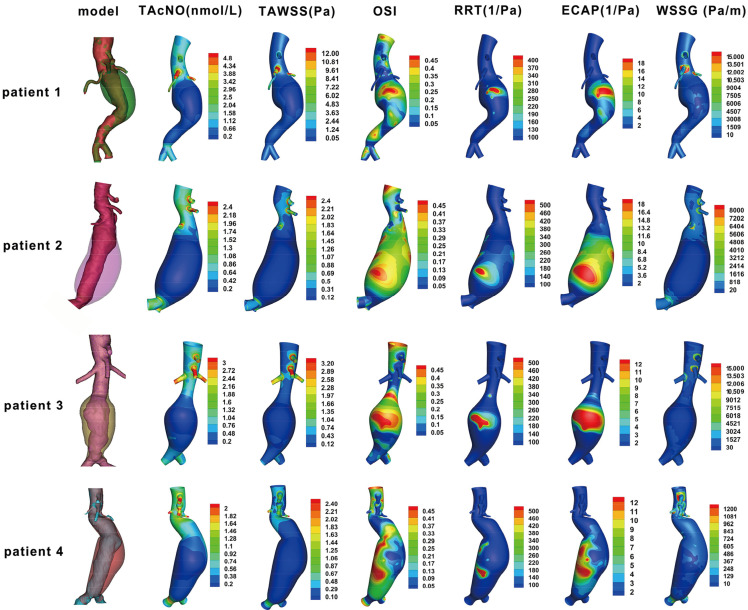
The geometry model, NO concentration, and CFD characteristics of 4 patients. TAcNO, time-average concentration of nitric oxide; TAWSS, time-average wall shear stress; OSI, oscillatory shear index; RRT, relative residence time; ECAP, endothelial cell activation potential; WSSG, wall shear stress gradient.

**Figure 3 bioengineering-12-00191-f003:**
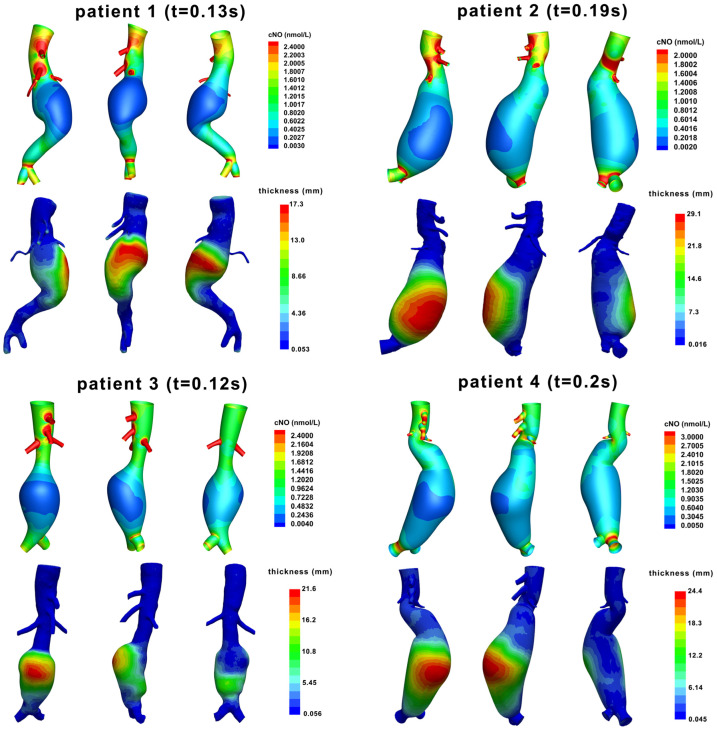
NO concentration at peak systole and ILT thickness. NO concentration at peak systole was shown for each patient. The thickness of thrombus was projected onto the outer wall.

**Figure 4 bioengineering-12-00191-f004:**
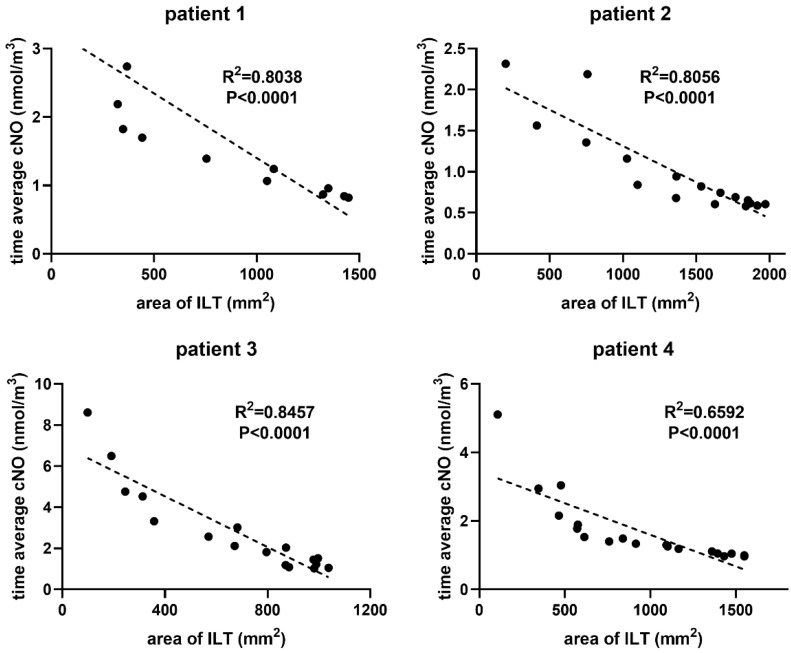
Scatter plot for the surface-averaged TAcNO concentration and the thrombus area of the ILT cross-sections.

**Figure 5 bioengineering-12-00191-f005:**
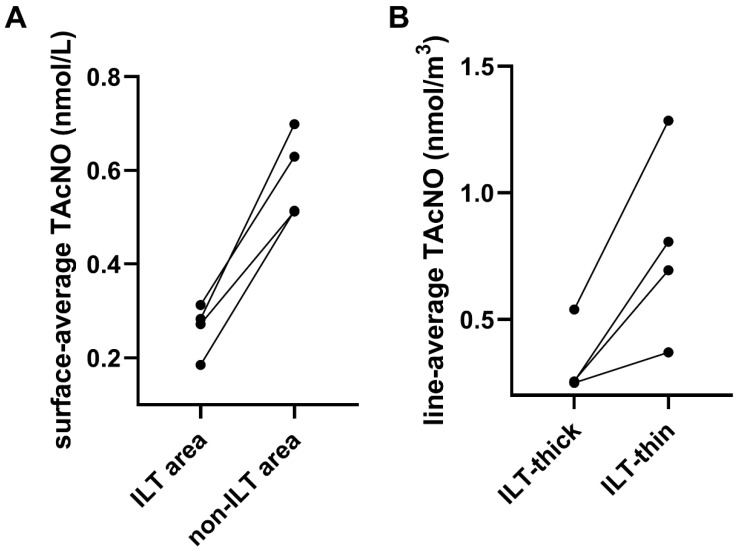
NO concentration between ILT and non-ILT sections or thin and thick ILT directions. (**A**) The average values of surface-average TAcNO values in the infra-renal aortic sections with thrombus (ILT) and without thrombus (non-ILT) were calculated for each patient. (**B**) The average values of line-average TAcNO in the thickest and thinnest ILT directions in three representative sections were calculated for each patient.

**Figure 6 bioengineering-12-00191-f006:**
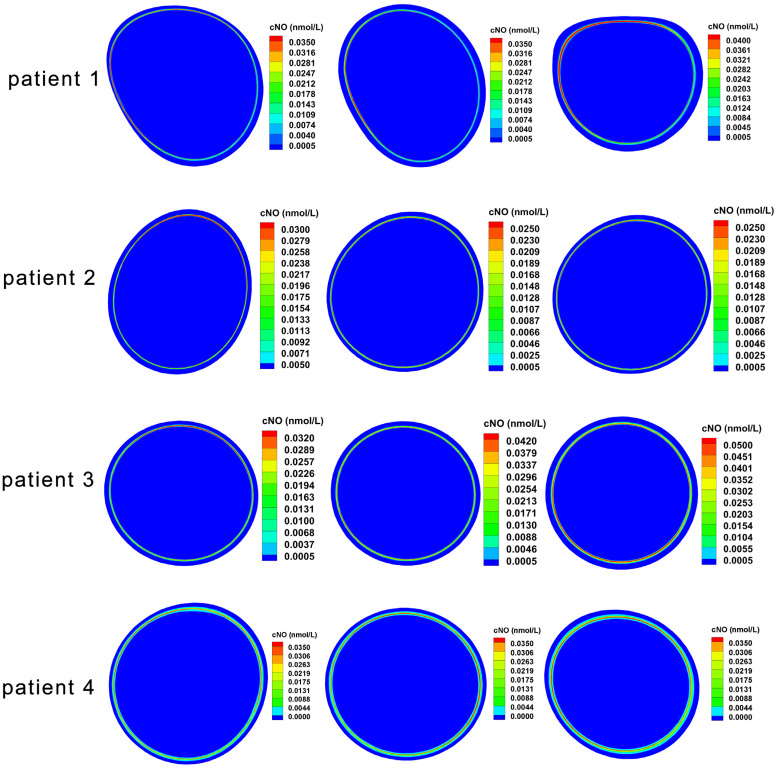
Surface-average TAcNO concentration in representative sections of ILT.

**Table 1 bioengineering-12-00191-t001:** Baseline information of four patients.

	*n* = 4
Age	63~80
Sex/male	3 (75)
Smoking	3 (75)
Hypertension	3 (75)
Hyperlipidemia	3 (75)
Coronary heart disease	2 (50)
NT-proBNP, pg/ml	72~104
Creatinine, μmol/L	74~93
GFR, mL/(min×1.73 m^2^)	70.86~88.29

Numbers were shown with (%) or range; GFR, glomerular filtration rate.

## Data Availability

The raw data supporting the conclusions of this article will be made available by the authors on request.

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
