# Peer review of "Nitric Oxide Distribution Correlates with Intraluminal Thrombus in Abdominal Aortic Aneurysm: A Computational Study"

_bioengineering, 2025, doi:10.3390/bioengineering12020191_

Round 1

Reviewer 1 Report

Comments and Suggestions for Authors

First and foremost, I would like to thank the editor for taking the time to review the submitted manuscript.

The study presents an investigation comparing NO production in the diagnosis of abdominal aortic aneurysm (AAA) and its correlation with the risk of intraluminal thrombus (ILT) formation. The manuscript is well-structured and includes all the necessary sections.

The main limitation of the study is the small sample size (n=4). I do not have experience with the methodology described by the authors and therefore cannot assess whether this sample size is sufficient for a study of this type. I would like to request the scientific editor’s evaluation regarding the adequacy of the sample size for this research.

I have no further objections to the manuscript.

Author Response

Comments: The study presents an investigation comparing NO production in the diagnosis of abdominal aortic aneurysm (AAA) and its correlation with the risk of intraluminal thrombus (ILT) formation. The manuscript is well-structured and includes all the necessary sections. The main limitation of the study is the small sample size (n=4). I do not have experience with the methodology described by the authors and therefore cannot assess whether this sample size is sufficient for a study of this type. I would like to request the scientific editor’s evaluation regarding the adequacy of the sample size for this research. 

Response: Thank you for your positive comments. Due to the time-consuming simulation process, we included a small sample size to explore the relationship between ILT and NO distribution. In future studies, we plan to expand the sample size to further validate our findings. This limitation has been addressed in the revised manuscript (Page 11, Line 345 in the revised manuscript).

Reviewer 2 Report

Comments and Suggestions for Authors

This manuscript presented an interesting simulated results for the effect of nitric oxide on the  intraluminal thrombus in the abdominal aortic aneurysm. The computational models were built based on CT images of patients with abdominal aortic aneurysm. CFD modelling was carried out. The equations of the transportation of NO was embedded through UDF. The results for 4 patients results were presented. The manuscript was overall written well but the introduction was only focusing on the general background for cardiovascular diseases rather than the effect of NO on the cardiovascular disease. The importance of the work will need to be emphasized. Therefore, it is suggested to re-write the introduction at least add more references and give more explanation for the correlation of the NO with vascular disease. 

The methodology only involved the general procedures of CFD but the key technology was nearly missed. The methodology will need to add more details regarding the implementation of UDF (transportation of NO).

The results were presented well with nice figures presented. However, interpretation of the results in depth such as the reason of the distribution of the NO and why it has an effect on wall shear stresses will be needed.

Author Response

Comments 1: This manuscript presented an interesting simulated results for the effect of nitric oxide on the  intraluminal thrombus in the abdominal aortic aneurysm. The computational models were built based on CT images of patients with abdominal aortic aneurysm. CFD modelling was carried out. The equations of the transportation of NO was embedded through UDF. The results for 4 patients results were presented. The manuscript was overall written well but the introduction was only focusing on the general background for cardiovascular diseases rather than the effect of NO on the cardiovascular disease. The importance of the work will need to be emphasized. Therefore, it is suggested to re-write the introduction at least add more references and give more explanation for the correlation of the NO with vascular disease. 

Response 1: Thank you for your valuable advice. The Introduction has been updated to include additional references on NO and vascular diseases (Page 2, Line 56 in the revised manuscript).

Comments 2:The methodology only involved the general procedures of CFD but the key technology was nearly missed. The methodology will need to add more details regarding the implementation of UDF (transportation of NO).

Response 2: Thank you for pointing this out. Based on your suggestion, we have revised the manuscript to provide a clearer explanation of the key technical aspects of UDF implementation (Page 4, Ling 134 in the revised manuscript). A detailed description has been added to explain how UDF was used to simulate NO transport and how the wall shear stress (WSS) obtained from flow field simulations was integrated into the NO transport model.

Comments 3: The results were presented well with nice figures presented. However, interpretation of the results in depth such as the reason of the distribution of the NO and why it has an effect on wall shear stresses will be needed.

Response 3: Thank you for your suggestion. In this study, NO distribution was modeled based  on three primary mechanisms: production, transport, and consumption. NO was produced by the endothelial layer and is moderated by local WSS. NO was transported within the lumen through a combination of diffusion and convection. NO was consumed by red blood cells (RBCs) (forming complex with hemoglobin)  and oxidated to nitrate by oxygen in the lumen. Therefore, both local WSS and overall flow conditions influenced NO distribution, making it a potential indicator of endothelial function and flow alterations caused by thrombus accumulation. Our results confirmed that NO distribution correlated more strongly with thrombotic regions compared to hemodynamic parameters such as WSS/WSSG, RRT, and ECAP. This discussion has been revised accordingly in the manuscript (Page 11, Line 318 in the revised manuscript).

Reviewer 3 Report

Comments and Suggestions for Authors

very nice paper

Author Response

Comments: very nice paper

Response: Thank you for your positive comments.

Round 2

Reviewer 2 Report

Comments and Suggestions for Authors

No further comments